# Out-of-Mold Sensor-Based Process Parameter Optimization and Adaptive Process Quality Control for Hot Runner Thin-Walled Injection-Molded Parts

**DOI:** 10.3390/polym16081057

**Published:** 2024-04-11

**Authors:** Feng-Jung Cheng, Chen-Hsiang Chang, Chien-Hung Wen, Sheng-Jye Hwang, Hsin-Shu Peng, Hsiao-Yeh Chu

**Affiliations:** 1Department of Mechanical Engineering, National Cheng Kung University, Tainan 701401, Taiwan; n16104131@gs.ncku.edu.tw (F.-J.C.); n16114089@gs.ncku.edu.tw (C.-H.C.); n18111011@gs.ncku.edu.tw (C.-H.W.); 2Department of Mechanical and Computer-Aided Engineering, Feng Chia University, Taichung 407102, Taiwan; hspeng@fcu.edu.tw; 3Department of Mechanical Engineering, Kun Shan University, Tainan 71070, Taiwan; hsiaoyeh@mail.ksu.edu.tw

**Keywords:** injection molding process, hot runner, thin-walled part, parameter optimization process, nozzle peak pressure, adaptive process control system

## Abstract

Injection molding is a highly nonlinear procedure that is easily influenced by various external factors, thereby affecting the stability of the product’s quality. High-speed injection molding is required for production due to the rapid cooling characteristics of thin-walled parts, leading to increased manufacturing complexity. Consequently, establishing appropriate process parameters for maintaining quality stability in long-term production is challenging. This study selected a hot runner mold with a thin wall fitted with two external sensors, a nozzle pressure sensor and a tie-bar strain gauge, to collect data regarding the nozzle peak pressure, the timing of peak pressure, the viscosity index, and the clamping force difference value. The product weight was defined as the quality indicator, and a standardized parameter optimization process was constructed, including injection speed, *V*/*P* switchover point, packing, and clamping force. Finally, the optimized process parameters were applied to the adaptive process control experiments using the developed control system operated within the micro-controller unit (MCU). The results revealed that the control system effectively stabilized the product weight variation and standard deviation of 0.677% and 0.0178 g, respectively.

## 1. Introduction

Distinct phenomena can be observed in the pressure curves at different stages of production during injection molding. The pressure curve characteristics that are highly correlated with product weight are defined as online quality indexes, which can serve as the basis for a standardized process parameter optimization procedure to stabilize product quality based on the relationship between pressure and specific volume. Given the above, it is clear that features’ different stages can reflect product quality and the process parameters of injection molding will directly impact these features. Therefore, optimizing process parameters is crucial for the performance and quality of the product.

The primary components of injection molding include materials, molds, and injection machines, and the equipment used for production is selected based on varying requirements. The utilization of hot runner systems can shorten production times, reduce injection pressures for the same components, and prove advantageous, helping to achieve cost savings and large-scale production [1]. Giboz et al. [2] investigated the size effect in the injection molding of thin-walled parts under various thickness ranges. The results indicate that a reduction in mold thickness leads to increased pressure and decreased volume flow rate, thereby increasing the complexity of the injection molding process. Multi-cavity molds can enhance production efficiency, where factors such as injection speed, mold temperature, and melt temperature influence the filling balance [3]. For thin-walled parts, parameters like injection speed and *V*/*P* switchover point are crucial for maintaining stable quality. Injection speed has more influence, since thinner products with thinner central layers are more susceptible to shear and thermal effects [4,5]. Yang et al. [6] mentioned that the viscosity of the plastic melt decreases with an increase in shear rate, and lower shear rates will lead to reductions in the product quality. For thin-walled products, reducing the melt viscosity is particularly crucial. Spina et al. [7] mentioned that thin-walled parts necessitate higher injection pressures and clamping forces, posing challenges in production. However, the effective use of hot runner systems could improve these issues. In summary, the combination of hot runner systems and hot gate molds can enhance production efficiency and stabilize product quality.

Many methods have been proposed for the optimization of process parameters based on real-time production data captured using sensors to monitor the injection molding process. Wang et al. [8] utilized cavity pressure sensors to capture data, plot pressure curves, and define the curve characteristics, indicating that peak pressure and the area under the curve are strongly correlated to product weight. Zhao et al. [9] proposed a method for online measurement of the clamping force using ultrasonic technology, demonstrating highly consistent results with traditional clamping-force-measuring instruments. Chen et al. [10] evaluated the quality of the melt by installing three pressure sensors in the nozzle, runner, and mold cavity, showing the strongest correlation between peak pressure and product quality, followed by the viscosity index calculated from pressure measurements at the nozzle and runner. Moayyedian et al. [11] applied the Taguchi method and fuzzy hierarchical analysis for a multi-objective optimization of injection molding to determine feasible molding performance indicators. Chen et al. [12] introduced a method to adjust the *V*/*P* switchover point using the elongation curve of a tie bar, showing that the characteristics of the clamping force increment curve were similar to those of the cavity pressure curve, making the method of determining the appropriate *V*/*P* switchover point feasible. Xu et al. [13] proposed a method based on the variation in clamping force to determine the optimal clamping force, showing that the optimal setting for the clamping force could be found when the clamping force variation was zero. Párizs et al. [14] proposed the feasibility of controlling the injection process using cavity pressure sensors in multi-cavity molds, reporting that the sensors could effectively detect the filling end time to determine the appropriate *V*/*P* switchover point, fit the pressure curve to determine the proper filling time, and analyze the pressure curve and pressure integral to determine the suitable clamping force. Huang et al. [15] found that setting the *V*/*P* switchover point too early or too late would significantly impact the cavity pressure curve and product quality. The utilization of multiple injection speeds could enhance product quality and reduce the final injection speed, helping to maintain consistent product quality. Kruppa et al. [16] proposed the use of a viscosity index to characterize the pressure curve, utilizing it as an indicator of product quality, and to determine the suitable *V*/*P* switchover point. Nian et al. [17] suggested that the optimal injection speed corresponds to the minimum cavity pressure integral, while identifying the suitable *V*/*P* switchover point entails aligning the screw position with the point of abrupt change in the cavity pressure profile.

The concept of adaptive control methods was introduced to ensure stable product quality over long-term production and mitigate the influence of the environment or different materials. This approach automatically adjusts parameters to stabilize production quality. Aminabadi et al. [18] utilized temperature and pressure sensors within the mold cavity, along with external mold clamping force strain gauges, to collect production data, predicting the next cycle’s product quality based on the sensor data. The machine automatically adjusted process parameters according to the prediction results to stabilize product quality in each cycle. Huang et al. [19] introduced an online real-time method that captured and fitted the cavity pressure curve in real-time during the filling stage to stabilize product quality, showing that online monitoring and curve fitting could effectively stabilize product quality due to environmental variations. Xu et al. [20] defined the time integral of pressure as the viscosity index and set it as the control factor. They achieved a stable product viscosity index through real-time, automatic adjustments of the *V*/*P* switchover point and filling pressure, thereby ensuring consistency in weight variations throughout long-term production. Schiffers et al. [21] suggested using the viscosity index as a means to determine the appropriate *V*/*P* switchover point, serving as a method for adaptive process control. Su et al. [22] installed two external sensors, namely a nozzle pressure sensor and a tie-bar strain gauge, to determine the appropriate *V*/*P* switchover point via the nozzle peak pressure and subsequently decide on the suitable clamping force using the clamping force peak value. Finally, the viscosity index and nozzle peak pressure are defined as the characteristic points that enable the monitoring of the *V*/*P* switchover point and injection speed in the adaptive process control system, while the clamping force difference value is used for flash monitoring. Liou et al. [23] recorded data from the nozzle pressure sensor and the tie-bar strain gauge, laying the foundation for parameter optimization, and the viability of this optimization process was then confirmed through experimentation with three polypropylene materials with different viscosities.

In the literature mentioned above, we found many studies on quality monitoring and parameter optimization in injection molding. However, there is relatively little research on parameter optimization and quality monitoring for hot runner and thin-walled products, despite the widespread applications of thin-walled products. Therefore, this study utilized hot runner thin-walled parts as the experimental product, installed a pressure sensor at the nozzle to capture nozzle pressure, and used a strain sensor placed on the tie bar to convert the instantaneous clamping force. By calculating the data captured by these two external sensors, the nozzle peak pressure, timing of peak pressure, viscosity index, and clamping force difference value were defined as online quality indexes. The process parameters, including injection speed, *V*/*P* switchover point, packing pressure, packing time, and clamping force, were sequentially optimized based on these online quality indexes. Finally, these process parameters were introduced into the developed adaptive process control system, which was based on a micro-controller unit (MCU) to conduct adaptive process control experiments to validate the effectiveness of the system in long-term production.

## 2. Methodology

### 2.1. Characteristics of the Nozzle Pressure Profile

In this study, the relationship between pressure and product weight was used to monitor the production process. Under a constant temperature, increasing the pressure will decrease the specific volume or increase the density. The final specific volume after cooling will significantly affect product quality, including the dimensions, mechanical properties, product weight, etc.

According to this relationship, the pressure generated in the injection process will affect the specific volume or density and directly affect the product weight. Therefore, set appropriate injection molding parameters to maintain a similar pressure curve for each cycle and the stability of the product weight will increase.

Therefore, the correlation between pressure and product weight is significant. In this study, the characteristics of the nozzle pressure curve, which highly correlated with product weight, were defined as online quality indexes. The previous literature demonstrated the effectiveness of fitting these characteristic points to stabilize product weight [22,23]. Furthermore, the injection molding parameters that highly correlated with product weight were adjusted to establish a standard pressure curve. In the study, the nozzle peak pressure, timing of peak pressure, and viscosity index were defined as the online quality indexes and were used to optimize injection speed, *V*/*P* switchover point, packing pressure, and packing time, as well as serving as the foundation for the adaptive process control system. Figure 1 shows the nozzle pressure profile characteristics.
Nozzle peak pressure (*P_peak_*): As the highest point on the nozzle pressure curve, the nozzle peak pressure occurs when the molten material transitions from flow to compression upon filling the mold, forming the peak of the nozzle pressure. This is utilized to optimize the injection speed and *V*/*P* switchover point, serving as an online quality index for the adaptive process control system.Timing of peak pressure (*t_peak_*): The instant of time when the nozzle peak pressure occurs is used to optimize the injection speed.Viscosity index (*VI*): The integral of pressure over time from the start of injection to the end of cooling is primarily employed as an online quality index for the adaptive process control system and assesses changes in product weight, as shown in Equation (1).
(1)$$VI=\int_{t_{injection\_start}}^{t_{cooling\_end}}P_{nozzle}(t)dt$$where *t_injection_start_* represents the time at the start of injection, *t_cooling_end_* represents the time at the end of cooling, and *P_nozzle_* represents the nozzle pressure.

### 2.2. Clamping Force Difference Value

The clamping force significantly influences the quality of the product, so appropriate parameter settings can stabilize the production process [24]. Higher injection speeds are required in production due to the rapid cooling characteristics of thin-walled products; thus, these products are more susceptible to mold separation, leading to flash. However, excessively high clamping force settings may result in poor venting and short shots and shorten the life of the mold and the injection molding machine.

In this study, a strain sensor was installed on the tie bar to monitor changes in the clamping force during injection. The clamping force difference value (∆*CF*) was defined as an online quality index to determine the appropriate clamping force to stabilize product quality and is the difference between the clamping force peak (*CF_peak_*) during injection and the clamping force set value (*CF_set_*), as shown in Equation (2) and Figure 2. *CF_peak_* represented the occurrence of flash, causing strain in the tie bar. The strain sensor received higher readings than usual, which were then converted into the clamping force peak.
(2)$$∆CF={CF}_{peak}-{CF}_{set}$$
where *CF_peak_* represents the clamping force peak and *CF_set_* represents the set value of the clamping force. A suitable clamping force is defined as the clamping force difference value of 0.

### 2.3. Adaptive Process Control System

To ensure the long-term production stability of the injection molding process, this research established an adaptive process control system within the MCU. The method of the adaptive process control system relies on the measurement of the nozzle peak pressure and viscosity index. However, the measured values can be influenced by environmental noise, leading to potential inaccuracies. To resolve this issue, this study employed a control strategy based on the average control chart theorem. In a stable production process, the nozzle peak pressure and the viscosity index should be distributed within the upper control limit (*UCL*) and the lower control limit (*LCL*). The limits of the *UCL* and *LCL* are set as two standard deviations. In each cycle, the system can calculate the characteristics of quality monitoring, automatically adjust the *V*/*P* switchover point and injection speed, and monitor the clamping force difference value.

The adaptive process control system adheres to the following strategy: (1) the injection speed will increase by 1 mm/s when the viscosity index exceeds the *UCL*; conversely, it will decrease by 1 mm/s. (2) The *V*/*P* switchover point will decrease by 0.1 mm when the nozzle peak pressure exceeds the *UCL*; conversely, it will increase by 0.1 mm. (3) Consistent monitoring of the clamping force difference value approaching 0 ensures that the injection process remains stable. In addition, once the system is activated, it will begin to capture data and perform system checks. Upon the official activation of the system for the second mold cycle, both online quality factor monitoring and process parameter adjustments will commence simultaneously. Figure 3 shows the control strategy of the adaptive process control system.

In the research, the weight of the injection molding product was defined as a measure of product quality. To ascertain the stability of the product weight during adaptive process control experiments, calculations were performed to determine both the variation and standard deviation of the product weight. A smaller value indicates greater stability.
(3)$$\mathrm{Variation}=\frac{W_{\max}-W_{\min}}{W_{\mathrm{average}}}\times 100\%$$
(4)$$\sigma =\sqrt{\frac{\sum_{i=1}^{N}{\left(x_{i}-\bar{x}\right)}^{2}}{N}}$$
where *Variation* represents the variation in the product weight, *W_max_* represents the maximum product weight, *W_min_* represents the minimum product weight, *W_average_* represents the average product weight, *σ* represents the standard deviation of the product weight, $\bar{x}$ represents the average product weight, and *N* represents the total number of products.

## 3. Experiment Setups

### 3.1. Materials

Polypropylene (PP) was utilized in this study and the material type was PP-BJ368MO (Borealis AG, Vienna, Austria). A highly flowable polypropylene with a melt flow index of 70 (g/10 min) was chosen as the experimental material to complement the high flow length ratio mold that was employed. The material properties are shown in Table 1.

### 3.2. Equipment

An all-electric injection molding machine (CLF-230AE, Chuan Lih Fa Co., Ltd., Tainan, Taiwan) was used, with a maximum injection rate of 398 cm^3^/s, maximum injection pressure of 1612 kg/cm^2^, and maximum clamping force of 230 tons. The hot runner mold with two cavities produced a cream box product with a thickness of 0.35 mm and a flow length ratio of 263.9, as shown in Figure 4. Figure 5 shows the experimental measurement system with a pressure sensor (PT4656XL, Dynisco, Franklin, MA, USA) installed on the nozzle to measure the pressure of the melt and a strain sensor (GE1029, Gefran, Provaglio d’Iseo, Italy) mounted on the tie bar to measure the strain. In this research, the sampling rate was 1000 Hz. Both external sensors were connected to a data acquisition system (DAQ USB-4716, Advantech Co., Ltd., Taipei, Taiwan) to collect data on nozzle pressure and clamping force. During the molding phase, the DAQ simultaneously gathered data from both sensors and the screw position from the control system, which were then imported for analysis. A single-chip microcomputer (AR-300T, ICP DAS Co. Ltd., Hsinchu, Taiwan) was used to replace the computer for the adaptive process control experiments. Each experiment was executed at least four times to ensure the correctness and reliability of the data.

## 4. Results and Discussion

### 4.1. Parameter Optimization Process

The conventional injection molding process relies heavily on operator experience for parameter selection and machine calibration. It often requires significant time and material resources to determine the optimal process parameters prior to production. To resolve this issue, previous studies have attempted to establish a standardized parameter optimization process for products of general thickness to achieve process parameter optimization and stabilize product weight using information from external sensors [23].

Considering the widespread applications of thin-walled products, this study focused on the parameter optimization of hot runner thin-walled parts. It established a parameter optimization process suitable for thin-walled products and observed the impact of different process parameters on product weight. The procedure was optimized by establishing a standard process parameter setup through the utilization of signals from sensors installed in the machine. This can find the appropriate process parameters without the need for experienced engineers.

In this research, the optimization experiment parameters were sequentially optimized. The parameter optimization process was primarily arranged according to the injection molding process. The first process parameter that was optimized was the injection speed, followed sequentially by the *V*/*P* switchover point, packing pressure, packing time, and clamping force. The optimized parameters in each experiment were used for the following optimization experiments. After the optimization of the above parameters, a set of close-to-optimal process parameters could be obtained. The parameters used for the optimization experiment are shown in Table 2.

#### 4.1.1. Injection Speed Experiments

Rapid cooling of the melt occurs during the injection molding of thin-walled parts and, as the product thickness decreases, the central layer of the melt becomes thinner, amplifying the influence of shear heating on the product [4,5]. Simultaneously, the cooling time becomes shorter, and an insufficient injection speed can lead to defects such as short shots. Therefore, injection speed is the primary parameter to be optimized for the injection molding of thin-walled parts.

Figure 6 shows the nozzle pressure profile at different injection speeds and Figure 7 shows the nozzle peak pressure and timing of nozzle peak pressure at different injection speeds. The nozzle peak pressure remained relatively stable at injection speeds below 175 mm/s, but the timing of the peak pressure decreased with increasing injection speed, indicating that the melt could fill the mold cavities more rapidly and smoothly, reducing the filling stage time. However, the shear heating effect increased melt flowability at injection speeds above 175 mm/s, causing the nozzle peak pressure to decrease. Nevertheless, the continued increase in injection speed raised melt temperature, increasing the specific volume and the peak pressure and leading to defects such as mold separation and flash, resulting in an unstable product weight.

As shown in Figure 8, there was an imbalance in the filling of two cavities, which improved with increasing injection speed. The results show that injection speeds from 125 mm/s to 175 mm/s contribute to maintaining stability in product weight. When the injection speed is too low, it leads to short shots, while an excessive injection speed causes mold separation, leading to an unstable product weight. The injection speed was set to 175 mm/s to prevent mold separation issues affecting product weight stability and to enhance production efficiency.

#### 4.1.2. *V*/*P* Switchover Point Experiments

The variations in nozzle pressure trends and product weight were observed by setting different *V*/*P* switchover points to define an appropriate *V*/*P* switchover point, with the setting of the injection speed based on the previous experiments. Figure 9 presents the nozzle pressure profile at different *V*/*P* switchover points, showing a significant increase in nozzle pressure when the melt transitions from flow to compression during the filling stage [23]. However, no pressure changes were observed. This was attributed to the use of a hot runner mold, which caused the pressure changes to not be transmitted to the nozzle sensor; therefore, the filling-to-compression transition was not observed. To address this issue, the appropriate *V*/*P* switchover point was defined by analyzing the nozzle peak pressure and product weight. Based on the variation in the nozzle peak pressure trend, the product transitions from short shot to filled and from filled to flash occurred at *V*/*P* switchover points between 13 mm and 15 mm. A further analysis of the product weight variation in different cavities revealed incomplete filling at 15 mm on the left cavity, whereas the right cavity was completely filled. At 14 mm, both cavities were completely filled without flash, as shown in Figure 10 and Figure 11. Therefore, considering the characteristics of thin-walled products, the *V*/*P* switchover point should be set after cavity filling, so the *V*/*P* switchover point was set at 14 mm.

#### 4.1.3. Packing Experiments

The nozzle pressure and product weight were changed through different packing pressures and time settings to determine the appropriate packing pressure and time using the injection speed and *V*/*P* switchover point from the previous experiments.

Figure 12 shows the nozzle pressure profiles at different packing pressures and Figure 13 shows nozzle pressure curves and screw positions for different packing pressures. The compensation of molten resin into the mold cavities was not smooth during the packing stage. The red arrows indicate the movement trends of the screw position at various packing pressures. As the packing pressure increased, the screw gradually moved forward to facilitate the compensation of molten resin into the mold cavities. The product weight gradually increased with an increase in packing time. However, the growth slowed after a packing time of 0.3 s, suggesting that the gate had frozen and could not compensate for the molten resin entering the cavities. This phenomenon occurred simultaneously in both cavities, as shown in Figure 14.

Furthermore, the setting of packing time and pressure parameters led to depression and deformation in the gate area of the product, particularly as the packing time and pressure increased. When molten plastic entered the mold cavities, it almost immediately cooled and solidified due to the fast cooling of thin-walled products. The packing pressure could only compensate for molten plastic near the gate, resulting in deformation. Therefore, the packing time and packing pressure settings were omitted.

#### 4.1.4. Clamping Force Experiments

The clamping force was optimized by analyzing the clamping force difference values at different clamping force settings, with a clamping force difference value of zero defined as the appropriate clamping force. The settings for injection speed, *V*/*P* switchover point, packing pressure, and packing time were based on the previous experiments.

Figure 15 presents the total weights and clamping force difference values at different clamping forces, showing that the clamping force difference value and product weight decreased as the clamping force increased. The appropriate clamping force was defined as 115 tons. The appropriate clamping force was multiplied by a safety factor of 1.2 to prevent instability during production leading to mold separation; thus, the clamping force was set at 138 tons.

### 4.2. Adaptive Process Control Experiments

To ensure the long-term production stability of the injection molding process, an adaptive process control system was established on the MCU, defining the nozzle peak pressure and viscosity index as online quality indexes. When online quality indexes exceed the *UCL* or *LCL*, the system provides parameter feedback to the machine, promptly adjusting the *V*/*P* switchover point or injection speed to ensure the stability of product weight. Simultaneously, it monitored clamping force information to ensure that no mold separation occurred during production. The control strategy of the adaptive process control system is shown in Figure 3. The adaptive process control experiments utilized the optimized parameters, conducting experiments with and without the adaptive control system for 100 cycles to validate the system. Table 3 shows the adaptive process control system’s experimental parameters.

Figure 16 shows the product weight with and without the adaptive process control system. The product weight was more stable when using the adaptive process control system, in both the left and right cavity. Additionally, the total product weight variation decreased from 0.819% to 0.677%, and the product weight standard deviation decreased from 0.02 g to 0.0178 g, confirming that the system effectively stabilized product weight, as shown in Table 4 and Table 5.

Figure 17 shows the nozzle peak pressure and viscosity. Figure 18 presents the adjustment process of the adaptive process control system. Starting from the second mold cycle, the system will monitor the online quality indexes and adjust the parameters. If the nozzle peak pressure and viscosity index exceed the upper or lower limits of the system setting, the system promptly provides feedback to adjust the parameter signals, thereby adjusting the parameters to stabilize the product weight.

Furthermore, the clamping force difference value remained at 0 throughout the experiments, indicating that the clamping force settings based on the optimized parameters are appropriate.

## 5. Conclusions

This study optimized the parameters for the production of hot runner thin-walled injection-molded parts, utilizing data collected from two external sensors, namely the nozzle pressure sensor and the tie-bar strain gauge, including the injection speed, *V*/*P* switchover point, packing, and clamping force. Subsequently, the optimized parameters were applied to an adaptive process control system to minimize product weight variations and standard deviation, successfully validating the effectiveness of the system. The study conclusions are as follows:An appropriate injection speed was defined based on the nozzle peak pressure, timing of the pressure peak, and product weight to reduce production time and ensure the stability of product weight during production.Since this study utilized a hot runner mold, the transition from the filling stage to the compression stage could not be observed, so the appropriate *V*/*P* switchover point was determined by observing the behavior of the melt through the nozzle peak pressure and product weight.The compensation effect of the melt during the packing stage for molded products was not ideal, with the packing pressure settings leading to deformations in the gate area, causing product defects.An appropriate clamping force was defined when the clamping force difference value was zero.The utilization of an adaptive control system effectively stabilized product weight, reducing weight variations and standard deviation from 0.819% and 0.02 g to 0.677% and 0.0178 g, respectively.

## Figures and Tables

**Figure 1 polymers-16-01057-f001:**
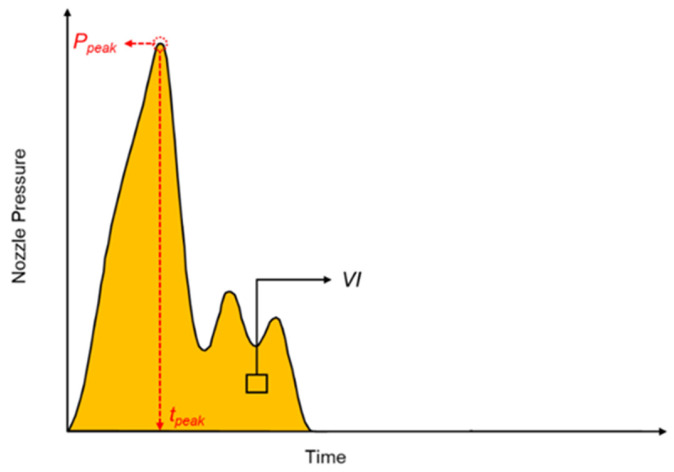
Nozzle pressure profile characteristics.

**Figure 2 polymers-16-01057-f002:**
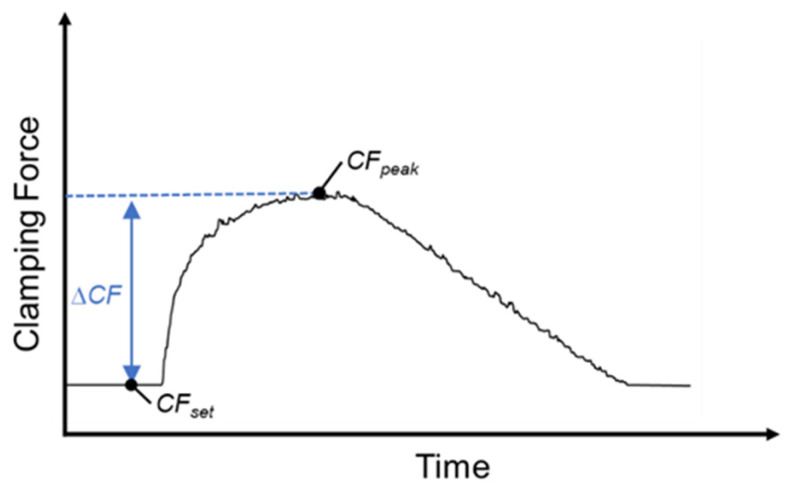
Clamping force difference value.

**Figure 3 polymers-16-01057-f003:**
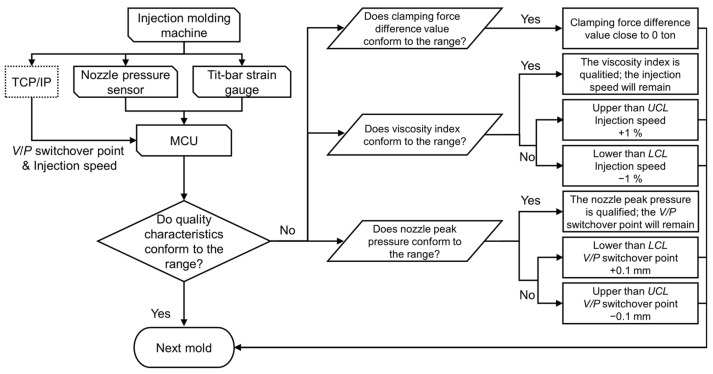
Control strategy of the adaptive process control system.

**Figure 4 polymers-16-01057-f004:**
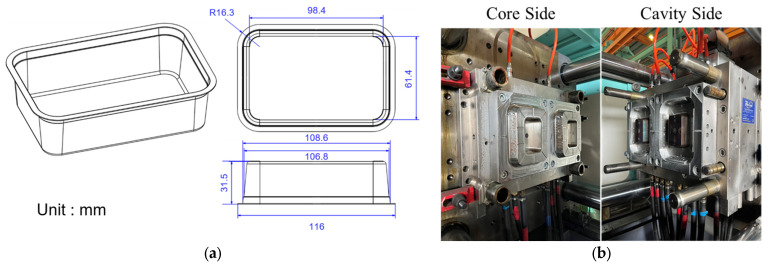
(**a**) Injection mold for the box sample and (**b**) the dimensions of the box sample.

**Figure 5 polymers-16-01057-f005:**
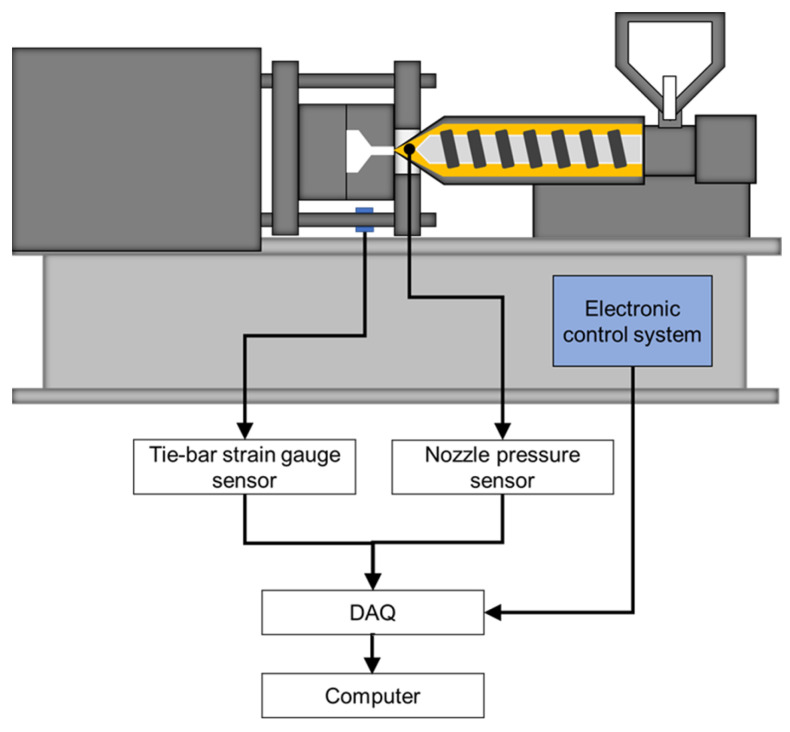
Experimental measurement system.

**Figure 6 polymers-16-01057-f006:**
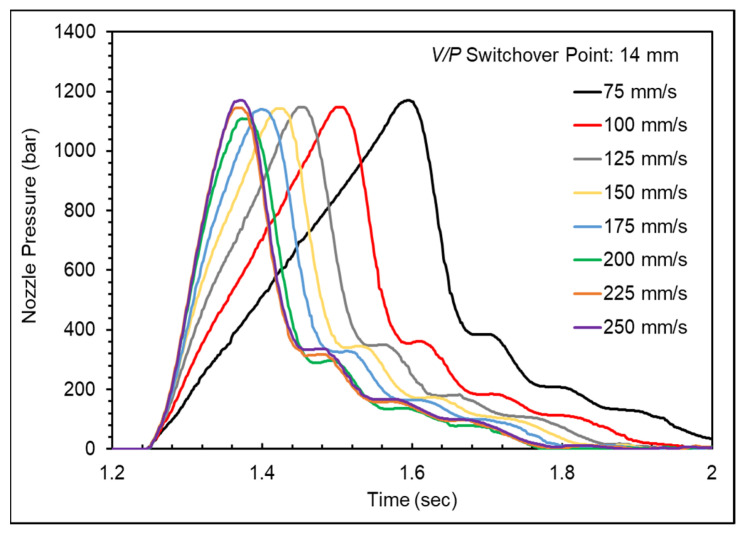
Nozzle pressure profile at various injection speeds.

**Figure 7 polymers-16-01057-f007:**
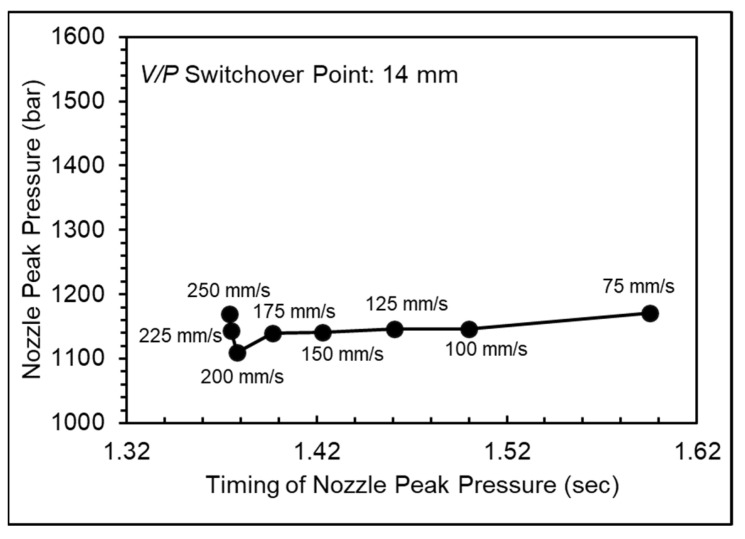
Nozzle peak pressure and timing of nozzle peak pressure at various injection speeds.

**Figure 8 polymers-16-01057-f008:**
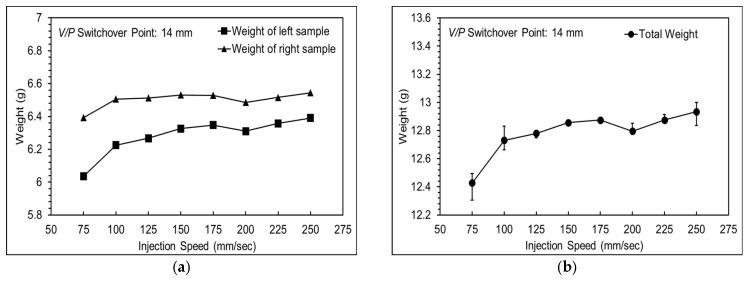
Product weights at various injection speeds, showing (**a**) the weight of each cavity, and (**b**) total weight.

**Figure 9 polymers-16-01057-f009:**
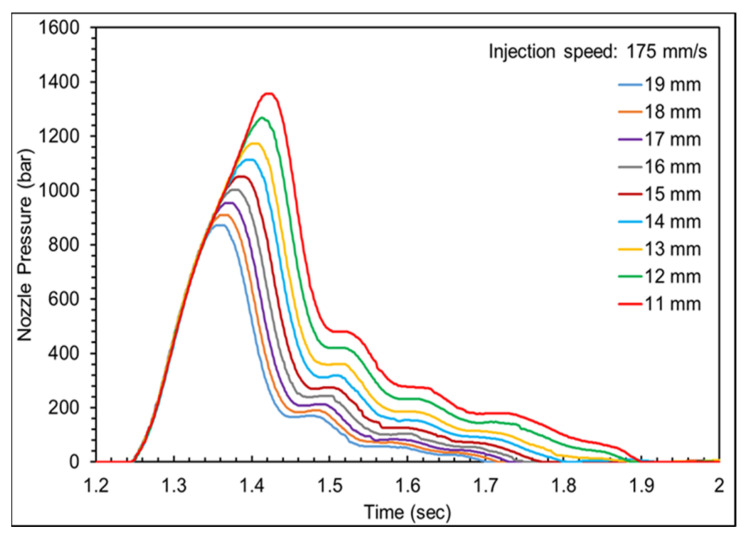
The nozzle pressure profile at various *V*/*P* switchover points.

**Figure 10 polymers-16-01057-f010:**
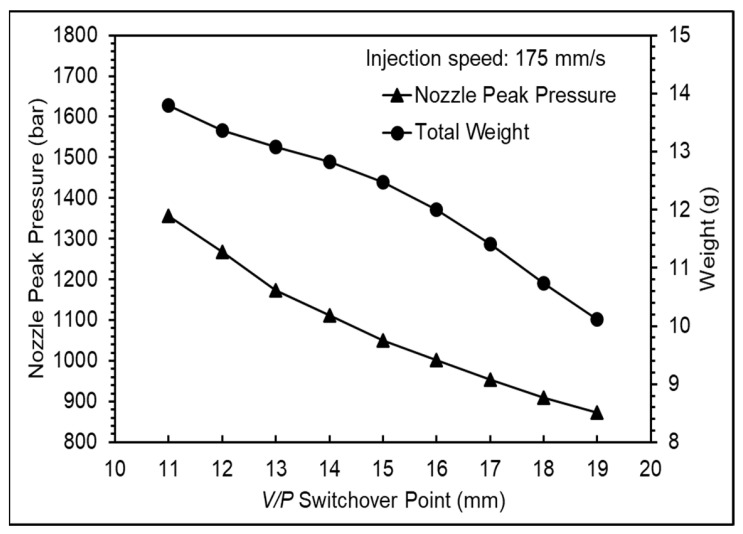
Nozzle peak pressure and total product weight at various *V*/*P* switchover points.

**Figure 11 polymers-16-01057-f011:**
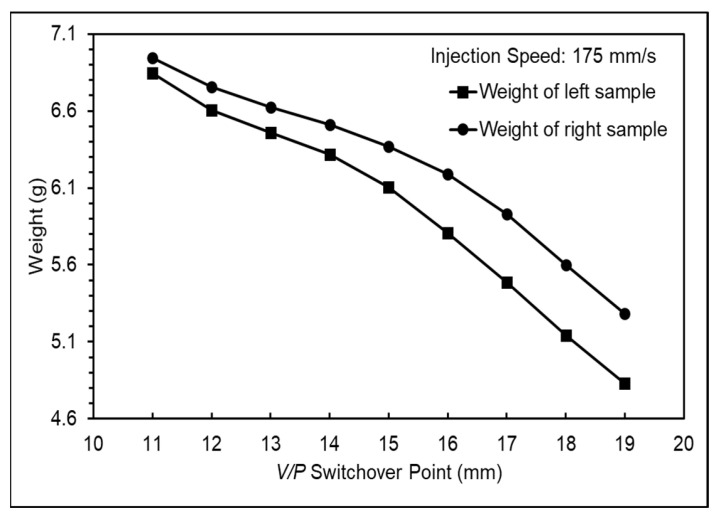
Product weight of each mold at various *V*/*P* switchover points.

**Figure 12 polymers-16-01057-f012:**
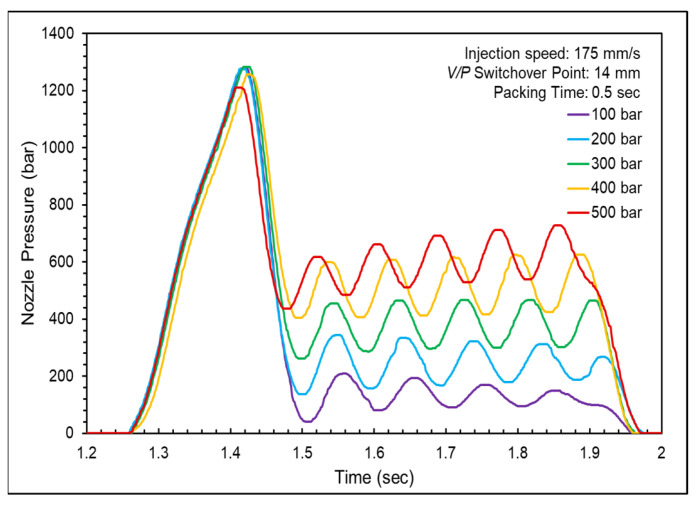
Nozzle pressure profiles at various packing pressures.

**Figure 13 polymers-16-01057-f013:**
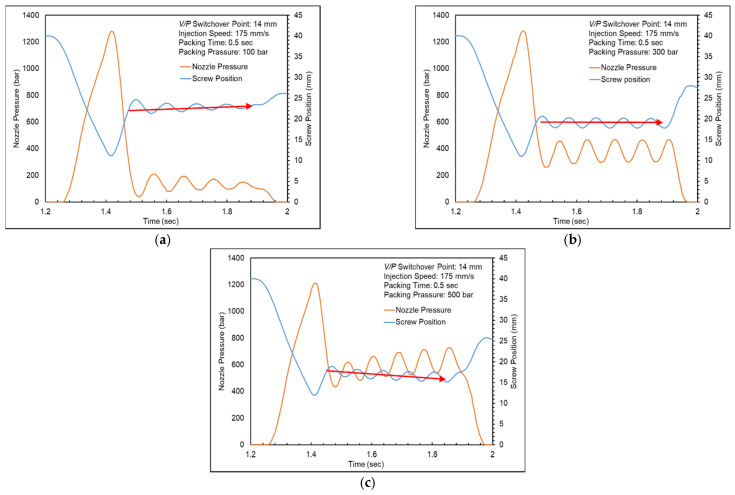
Nozzle pressure curves and screw positions for packing pressures of (**a**) 100 bar, (**b**) 300 bar, and (**c**) 500 bar.

**Figure 14 polymers-16-01057-f014:**
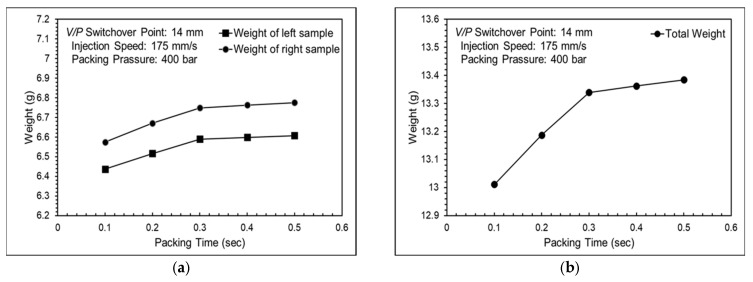
Product weights with a packing pressure of 400 bar, showing results (**a**) for each cavity and (**b**) total weight.

**Figure 15 polymers-16-01057-f015:**
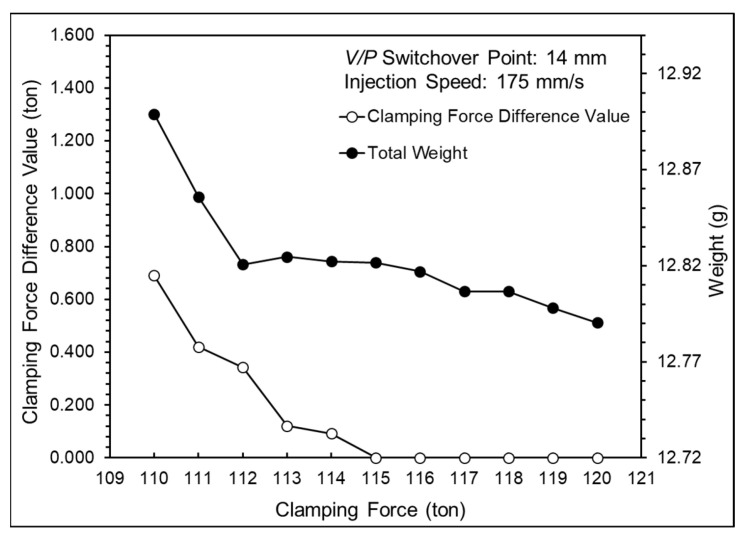
Total weights and clamping force difference values at various clamping forces.

**Figure 16 polymers-16-01057-f016:**
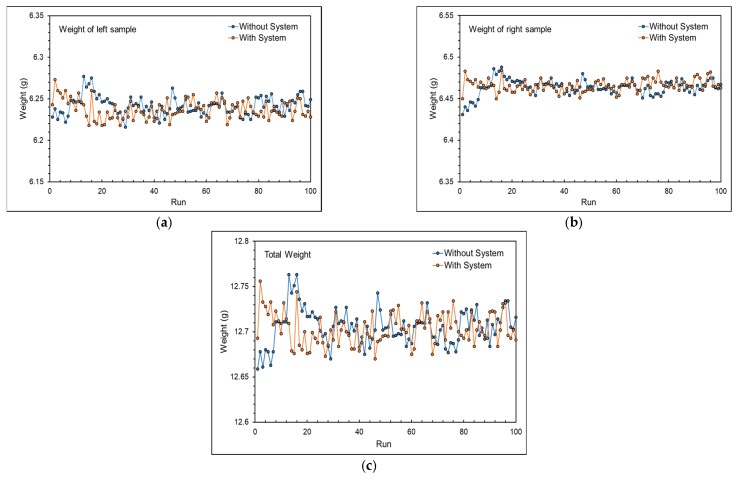
Product weight with and without the adaptive process control system: (**a**) left sample, (**b**) right sample, and (**c**) total.

**Figure 17 polymers-16-01057-f017:**
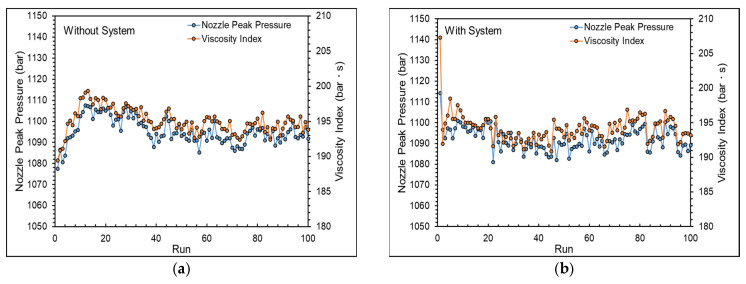
Nozzle peak pressure and viscosity (**a**) without the system and (**b**) with the system.

**Figure 18 polymers-16-01057-f018:**
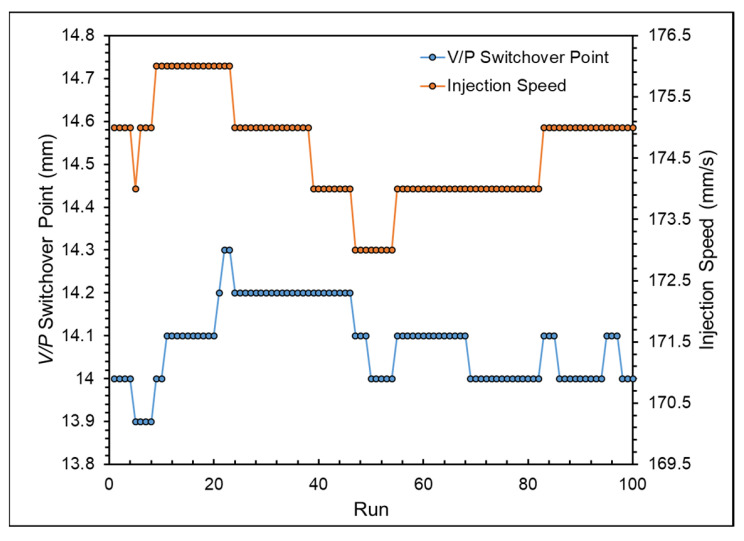
Adjustment of the adaptive process control system.

**Table 1 polymers-16-01057-t001:** Material properties.

BJM368MO	Unit	Value
Melt Flow Index	g/10 min	70
Density	g/cm^3^	0.905
Shrinkage	%	1.2

**Table 2 polymers-16-01057-t002:** The parameters for the optimization experiments.

Initial Parameters			
Injection pressure (bar)	1580	Cooling time (s)	3
Melt temperature (°C)	230	*V*/*P* switchover point (mm)	14
Mold temperature (°C)	25	Packing time (s)	0
Hot runner temperature (°C)	230	Clamping force (ton)	166
**Optimization Experiment Parameters**			
Injection speed (mm/s)	75, 100, 125, 150, 175, 200, 225, 250		
*V*/*P* switchover point (mm)	11, 12, 13, 14, 15, 16, 17, 18, 19		
Packing pressure (bar)	100, 200, 300, 400, 500		
Packing time (s)	0.1, 0.2, 0.3, 0.4, 0.5		
Clamping force (ton)	110, 111, 112, 113, 114, 115, 116, 117, 118, 119, 120		

**Table 3 polymers-16-01057-t003:** Adaptive process control system’s experimental parameters.

Experimental Parameters			
Injection pressure (bar)	1580	Cooling time (s)	3
Melt temperature (°C)	230	Injection speed (mm/s)	175
Mold temperature (°C)	25	*V*/*P* switchover point (mm)	14
Hot runner temperature (°C)	230	Packing time (s)	0
Clamping force (ton)	138		

**Table 4 polymers-16-01057-t004:** Variation in the product weight.

Weight	Without System (%)	With System (%)
Left sample	0.977	0.882
Right sample	0.882	0.526
Total	0.819	0.677

**Table 5 polymers-16-01057-t005:** Standard deviation of the product weight.

Weight	Without System (g)	With System (g)
Left sample	0.0113	0.0112
Right sample	0.0095	0.0073
Total	0.02	0.0178

## Data Availability

The data presented in this study are available on request from the corresponding author.

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
