# Peer review of "Out-of-Mold Sensor-Based Process Parameter Optimization and Adaptive Process Quality Control for Hot Runner Thin-Walled Injection-Molded Parts"

_polymers, 2024, doi:10.3390/polym16081057_

Round 1
Reviewer 1 Report (Previous Reviewer 2)
Comments and Suggestions for Authors
Dear Authors,
Thank you for your response and for providing a detailed revision and point-by-point response to the reviewer’s feedback. After reviewing your revisions, it is evident that you have made significant improvements to the manuscript by addressing the concerns raised by the reviewer.
You have incorporated the necessary changes, including providing a more sufficient and quantitative theoretical basis for the parameter selection method, clarifying the definition of the viscosity index, and providing a detailed theoretical demonstration for the adjustment relationship in the adaptive control system strategy. These revisions have strengthened the scientific rigor and clarity of your study. And after reviewing the manuscript, I also found some minor details, which I hope will attract your attention:
1. The experimental measurement system should provide specific physical measurement to confirm the authenticity and reliability of the system in figure 5
2. There is a high degree of similarity between the research methods used in this manuscript and in reference 22, and the differences between them should be pointed out.
Considering the efforts you have made in revising the manuscript and the improvements made based on the reviewer’s feedback, we are pleased to inform you that the manuscript is now suitable for publication in a polymer journal. We appreciate your dedication and hard work in addressing the reviewer’s comments. Congratulations on the improved manuscript, and we wish you success in the publication process.
Author Response
Response to Reviewer 1 Comments
<Polymers>
< Out-of-Mold Sensor Based Process Parameter Optimization and Adaptive Process Quality Control for Hot Runner Thin-walled Injection Molded Parts >
<Manuscript ID: polymers- 2956237>
Dear Reviewer,
We appreciate the time and effort the reviewer has dedicated to providing insightful feedback on ways to strengthen our paper. Thus, it is with great pleasure that we were doing our best to work on revising our article for further consideration. We have incorporated changes that reflect the detailed suggestions you have graciously provided. We also hope that our efforts and responses in revising the paper as shown below will satisfactorily address all the issues and concerns you and the reviewers have noted.
Point 1.
The experimental measurement system should provide specific physical measurement to confirm the authenticity and reliability of the system in figure 5.
Response 1.
The authors have added the details of the measurement system (line 233-240). The description is as follows:
Figure 5 shows the experimental measurement system with a pressure sensor (PT4656XL, Dynisco, USA) installed on the nozzle to measure the pressure of the melt and a strain sensor (GE1029, Gefran, Italy) mounted on the tie bar to measure the strain. In this research, the sampling rate was 1000 Hz. Both external sensors were connected to a data acquisition system (DAQ USB-4716, Advantech Co., Ltd., Taiwan) to collect data on nozzle pressure and clamping force. During the molding phase, the DAQ simultaneously gathered data from both sensors and the screw position from the control system, which were then imported for analysis. A single-chip microcomputer (AR-300T, ICP DAS Co. Ltd., Taiwan) was used to replace the computer for the adaptive process control experiments. Each experiment was executed at least four times to make sure the correctness and reliability of the data.
Point 2.
There is a high degree of similarity between the research methods used in this manuscript and in reference 22, and the differences between them should be pointed out.
Response 2.
The authors have pointed out the differences between this paper and Reference 22:
- This study conducted injection experiments using hot runner thin-walled parts with flow length of 263.9. According to the literature, rapid cooling of the melt occurs during the injection molding process of thin-walled parts, high injection speeds were necessary for experimentation. However, there is relatively little research on parameter optimization and quality monitoring for hot runner and thin-walled parts, despite the widespread applications of thin-walled products. Therefore, this study primarily analyzed the variation in nozzle pressure and tie-bar strain under high-speed injection molded thin-walled parts, identifying characteristic points for parameter optimization process. The injection molded part fabricated in Reference 22 is a 2 mm part which is a part with normal thickness or flow length of 42.3.
- In Reference 22, sequential experiments were conducted with short-shot experiments, appropriate clamping force experiments, and full-factorial experiments. These experiments analyze the influence of injection speed, V/P switchover point, and clamping force on the nozzle pressure profile and clamping force. However, packing time or packing pressure setup were not included in the paper of Reference 22. Therefore, basing on the experimental results of Reference 22, this study refined the experiment methodology with a more systematic approach to setup the process parameters. A standard parameter optimization process was established based on the nozzle pressure profile and clamping force data. Optimization experiments were sequentially conducted for injection speed, V/P switchover point, packing pressure, packing time, and clamping force, aiming to determine a set of suitable process parameters.
We appreciate your kind help and hope to hear from you soon.
Sincerely,
Sheng-Jye Hwang
Professor
Department of Mechanical Engineering
National Cheng Kung University
Tainan, Taiwan

Reviewer 2 Report (Previous Reviewer 3)
Comments and Suggestions for Authors
It can be accepted.
Author Response
Response to Reviewer 2 Comments
<Polymers>
< Out-of-Mold Sensor Based Process Parameter Optimization and Adaptive Process Quality Control for Hot Runner Thin-walled Injection Molded Parts >
<Manuscript ID: polymers-2956237>
Dear Reviewer,
Thanks very much for your kind work and consideration on publication of our paper. On behalf of my co-authors, we would like to express our great appreciation to editor and reviewers.
Thank you and best regards.
Sincerely,
Sheng-Jye Hwang
Professor
Department of Mechanical Engineering
National Cheng Kung University
Tainan, Taiwan
This manuscript is a resubmission of an earlier submission. The following is a list of the peer review reports and author responses from that submission.
Round 1
Reviewer 1 Report
Comments and Suggestions for Authors
polymers-2850899
Out-of-Mold Sensor Based Process Parameter Optimization and Adaptive Process Quality Control for Hot-Runner Thin-Wall Injection Molded Parts
Abstract, Section 1 Introduction and Section 2 Methodology – “Product quality” (an output) is used without a definition, except a passing reference to product weight. If product weight is specifically meant to be product quality, then please replace “product quality”. If product quality means weight and other product characteristics, please define them early in the manuscript. Section 2.4 and equations 3-4 are quite specific that quality is part weight.
Top of Page 4, “ … product weight from the nozzle pressure curve and defining them as online quality indexes “ is very confusing. “In this study, the nozzle peak pressure, timing of peak pressure, and viscosity index were defined as the online quality indexes and were used to optimize injection speed, V/P switchover point, packing pressure, and packing time, as well as serve as the foundation for the adaptive process control system.” What are the inputs and outputs being referred to when referring to “product quality” and “quality indexes”?
Page 4 How is equation 1 related to viscosity?
Page 4 Equations 3 and 4 – why use the variability (range) and standard deviation rather than the target part weight? With the definitions used there could be part weight drift with an underlying long term disturbance, such as injection mold cooling fouling.
Page 13 and 14, Figures 18 and 19 How are the two inputs, tie-bar strain gauge sensor and nozzle pressure sensor, used in MCU algorithm to the injection speed and V/P switchover point? Figure 19 seems to imply small independent incremental moves in the two variables, but I don’t understand how they are connected to the clamping force, viscosity index and nozzle peak pressure. Very confusing.
Page 15 Figures 20 and 21 Is there any quantitative comparison of the two time series? On inspection, the variability does not look very different between the cases with and without the adaptive control system.
Recommended reference: Injection Molding Process Control, Monitoring, and Optimization, Hanser Publishers, Yang, Y., Chen, X., Lu, N., Gao, F. ISBN: 978-1-56990-593-7 2016
Comments on the Quality of English LanguageSome editing required so that the language is more specific and precise in meaning.
Author Response
Response to Reviewer 1 Comments
<Polymers>
< Out-of-Mold Sensor Based Process Parameter Optimization and Adaptive Process Quality Control for Hot-Runner Thin-walled Injection Molded Parts>
<Manuscript ID: polymers-2850899>
Dear Reviewer,
We appreciate the time and effort the reviewer has dedicated to providing insightful feedback on ways to strengthen our paper. Thus, it is with great pleasure that we were doing our best to work on revising our article for further consideration. We have incorporated changes that reflect the detailed suggestions you have graciously provided. We also hope that our efforts and responses in revising the paper as shown below will satisfactorily address all the issues and concerns you and the reviewers have noted.
Point 1.
Abstract, Section 1 Introduction and Section 2 Methodology – “Product quality” (an output) is used without a definition, except a passing reference to product weight. If product weight is specifically meant to be product quality, then please replace “product quality”. If product quality means weight and other product characteristics, please define them early in the manuscript. Section 2.4 and equations 3-4 are quite specific that quality is part weight.
Response 1.
The authors have revised certain definitions in the revised manuscript to make it clearer. The description is as follows:
Previous literature demonstrated the effectiveness of fitting these characteristic points for stabilizing product weight [22-23]. (line 145-147)
Viscosity index (VI): The integral of pressure over time from the start of injection to the end of cooling is primarily employed as an online quality index for the adaptive process control system and assesses changes in product weight, as shown in Equation 1. (line 160-163)
However, excessively high clamping force settings may result in poor venting, short shots and shortens the life of the mold and the injection molding machine. (line 172-174)
In the research, the weight of the injection molding product was defined as a measure of product quality. In order to determine the stability of the product weight during adaptive process control experiments, the variation and standard deviation of the product weight were calculated. (line 207-210)
It established a parameter optimization process suitable for thin-walled products and observing the impact of different process parameters on product weight. (line 250-252)
The utilization of an adaptive control system effectively stabilized product weight, reducing weight variation and standard deviation from 0.819% and 0.02 g to 0.677% and 0.0178 g, respectively. (line 410-412)
Point 2.
Top of Page 4, “ … product weight from the nozzle pressure curve and defining them as online quality indexes “ is very confusing. “In this study, the nozzle peak pressure, timing of peak pressure, and viscosity index were defined as the online quality indexes and were used to optimize injection speed, V/P switchover point, packing pressure, and packing time, as well as serve as the foundation for the adaptive process control system.” What are the inputs and outputs being referred to when referring to “product quality” and “quality indexes”?
Response 2.
- The description in Section 2.2 has been modified (line 143-152). The description is as follows:
According to the P-V-T relationship, the correlation between pressure and product weight is significant. In the study, the characteristics of the nozzle pressure curve which highly correlated with product weight were defined as online quality indexes. Previous literature demonstrated the effectiveness of fitting these characteristic points for stabilizing product weight [22-23]. Furthermore, the injection molding parameters highly correlated with product weight were adjusted to establish a standard pressure curve. In the study, the nozzle peak pressure, timing of peak pressure, and viscosity index were defined as the online quality indexes and were used to optimize injection speed, V/P switchover point, packing pressure, and packing time, as well as serve as the foundation for the adaptive process control system.
- In the study, the product weight is defined as a measure of product quality. And the online quality indexes are calculated from data captured by sensors, allowing for a quick determine of product quality without measuring product weight.
Point 3.
Page 4 How is equation 1 related to viscosity?
Response 3.
The viscosity index is defined as the integral of the pressure curve with respect to time. Due to its final unit being the same as that of viscosity, it is referred to as the viscosity index. According to the literature, the viscosity index is highly correlated with product weight. Therefore, in this study, by monitoring the viscosity index, the injection speed is adjusted to stabilize the product weight during long-term production. The authors have already incorporated the literature on the viscosity index into the article to improve the content of theory. The description is as follows:
Kruppa [16] et al. proposed the viscosity index to characterize the pressure curve, utilizing it as an indicator of product quality and to determine the suitable V/P switchover point. (line 82-84)
Chen [17] et al. verified that changes in the viscosity index, which is the pressure–time integral. The results show that the highly correlated among viscosity index, V/P switchover point and product weight. (line 84-86)
Point 4.
Page 4 Equations 3 and 4 – why use the variability (range) and standard deviation rather than the target part weight? With the definitions used there could be part weight drift with an underlying long term disturbance, such as injection mold cooling fouling.
Response 4.
In this research, the adaptive process control system aims to reduce the instability in the continuous process. The variation and standard deviation of the product weight are defined to measure the stability in the production process. Due to environmental noise such as material batch variations and temperature, defining the target part weight is challenging. In the future, the authors will attempt to incorporate the target part weight into the research to achieve higher-quality product. Thank you very much for your suggestion.
Point 5.
Page 13 and 14, Figures 18 and 19. How are the two inputs, tie-bar strain gauge sensor and nozzle pressure sensor, used in MCU algorithm to the injection speed and V/P switchover point? Figure 19 seems to imply small independent incremental moves in the two variables, but I don’t understand how they are connected to the clamping force, viscosity index and nozzle peak pressure. Very confusing.
Response 5.
The authors have added the description control strategy of adaptive process control system and modified the flow chart (line 187-206). The revision is as follows:
To ensure the long-term production stability of the injection molding process, this research established an adaptive process control system within the MCU. The method of adaptive process control system relies on the measurement of the nozzle peak pressure and viscosity index. However, the measured values can be influenced by environmental noise, leading to potential inaccuracies. To resolve this issue, this study employed a control strategy based on average control chart theorem. In a stable production process, the nozzle peak pressure and the viscosity index should distribute within the upper control limit (UCL) and lower control limit (LCL). In each cycle, the system can calculate the characteristics of quality monitoring and automatically adjust the V/P switchover point and injection speed and monitor clamping force difference value.
The adaptive process control system adheres to the following strategy: (1) the injection speed will increase by 1 mm/s when the viscosity index exceeds the UCL; conversely, it will decrease 1 mm/s. (2) the V/P switchover point will decrease by 0.1 mm when the nozzle peak pressure exceeds the UCL; conversely, it will increase 0.1 mm. (3) consistent monitoring of the clamping force difference value approaching 0 ensures that the injection process remains stable. Figure 3 shows the control strategy of the adaptive process control system.
Figure 3. Control strategy of the adaptive process control system.
Point 6.
Page 15 Figures 20 and 21. Is there any quantitative comparison of the two time series? On inspection, the variability does not look very different between the cases with and without the adaptive control system.
Response 6.
The authors have adjusted the scale of product weight range of Figures 20 and 21 (Figure 16 of the revised manuscript) to make the data clearer.
- In the research, the automatic mode was employed for the adaptive process control experiments, ensuring that the time spent in each cycle remains the same.
- The phenomenon of a slight reduction in variation and standard deviation is attributed to the optimization of process parameters conducted in the parameter optimization experiments, and the application of these optimized parameters in the adaptive control experiments. As the result, the product weight has become more stable. However, even with close-to-optimal parameters, product weight may still be influenced by environmental noise. Therefore, the primary objective of the adaptive process control system is to further stabilize the overall production process.
Point 7.
Recommended reference: Injection Molding Process Control, Monitoring, and Optimization, Hanser Publishers, Yang, Y., Chen, X., Lu, N., Gao, F. ISBN: 978-1-56990-593-7 2016.
Response 7.
The relevant reference has been included in this study. The description is as follows:
Yang [6] et al. mentioned that the viscosity of plastic melt decreases with an increase in shear rate, and lower shear rates will lead to defects in the product quality. For thin-walled products, reducing melt viscosity is particularly crucial. (line 48-51)
We appreciate your kind help and hope to hear from you soon.
Sincerely,
Sheng-Jye Hwang
Professor
Department of Mechanical Engineering
National Cheng Kung University
Tainan, Taiwan

Reviewer 2 Report
Comments and Suggestions for Authors
This manuscript addresses the challenges of maintaining quality stability in long-term production of thin-wall parts in injection molding and provides valuable insights for optimizing process parameters and improving product quality in the injection molding industry.I think this fits within the scope of the journal Polymers. Specific comments and suggestions can be found in word.

Comments on the Quality of English LanguageThe English language quality of this manuscript is overall good. The language is clear and effectively conveys the technical information related to injection molding. The sentences are well-structured and flow smoothly, making it easy to understand the key points being discussed. The use of appropriate vocabulary and terminology adds to the clarity of the text. Additionally, the information is presented in a logical sequence, allowing the reader to follow the process and understand the study’s objectives and findings. There are no grammatical errors or major issues with sentence structure. Overall, this manuscript demonstrates a good command of the English language and effectively presents the technical information related to the study.
Author Response
Response to Reviewer 2 Comments
<Polymers>
< Out-of-Mold Sensor Based Process Parameter Optimization and Adaptive Process Quality Control for Hot-Runner Thin-walled Injection Molded Parts>
<Manuscript ID: polymers-2850899>
Dear Reviewer,
We appreciate the time and effort the reviewer has dedicated to providing insightful feedback on ways to strengthen our paper. Thus, it is with great pleasure that we were doing our best to work on revising our article for further consideration. We have incorporated changes that reflect the detailed suggestions you have graciously provided. We also hope that our efforts and responses in revising the paper as shown below will satisfactorily address all the issues and concerns you and the reviewers have noted.
Point 1.
The Rationality of parameter optimization method: For the selected injection speed, V/P switching point and clamping force and other process parameters in table 2, the optimized parameters obtained are obtained from several sets of process parameters set by the author. Is there a more sufficient and quantitative theoretical basis to explain the rationality and effectiveness of parameter selection?
Response 1.
The parameter optimization method in this study was determined based on previous results and relevant literatures. The following are the pertinent references:
Su et al. [22] installed two external sensors, namely a nozzle pressure sensor and a tie-bar strain gauge. Determining the appropriate V/P switchover point by the nozzle peak pressure and subsequently deciding the suitable clamping force by the clamping force peak value. Finally, defining the viscosity index and nozzle peak pressure as the characteristic points enables the monitoring of the V/P switchover point and injection speed in the adaptive process control system, while the clamping force difference value is used for flash monitoring. (line 104-110)
Liou et al. [23] recorded data from the nozzle pressure sensor and the tie-bar strain gauge, laying the foundation for parameter optimization and the viability of this optimization process was then confirmed through experimentation with three polypropylene materials with different viscosity. (line 110-113)
Wang et al. [8] utilized cavity pressure sensors to capture data, plot pressure curves, and defined the curve characteristics, indicating that peak pressure and the area under the curve were strongly correlated to product weight. (line 58-60)
Chen et al. [10] evaluated the quality of the melt by installing three pressure sensors in the nozzle, runner, and mold cavity, showing the strongest correlation between peak pressure and product quality, followed by the viscosity index calculated from pressure measurements at the nozzle and runner. (line 62-66)
Xu et al. [13] proposed a method based on the variation of clamping force to determine the optimal clamping force showing that the optimal setting for the clamping force could be found when the clamping force variation was zero. (line 71-74)
Huang et al. [15] found that setting the V/P switchover point too early or too late would significantly impact the cavity pressure curve and product quality. The utilization of multiple injection speeds could enhance product quality and reduce the final injection speed, maintaining consistent product quality. (line 79-82)
Several studies utilize the cavity sensors to measure injection molding process data. The objective of this research is to install sensors in the nozzle to monitor the injection molding process, optimizing injection molding parameters, and stabilizing the product weight under reasonable cost. Furthermore, there is relatively little research on parameter optimization and quality monitoring for hot runner and thin-walled products, despite the widespread applications of thin-walled products. Therefore, this study selected hot runner thin-walled parts as the experimental product.
Point 2.
Comparison of parameter optimization methods: Consider comparing the selected parameter optimization method with other commonly used optimization methods, such as response surface method, genetic algorithm, etc. This will help to assess the strengths and limitations of the proposed methods and further enhance the reliability and comparability of the studies.
Response 2.
This study mentioned the parameter optimization method based on the curve characteristics collected by sensors. Since the method is still in the experimental stage, and no other optimization methods have been included for comparison. In future research, the authors will consider adding alternative optimization methods to evaluate the strengths and limitations of the proposed optimization method.
Point 3.
Accuracy of the definition of viscosity index (VI) : Combing the concepts of injection start time, cooling start time and cooling end time in equation 1.
Response 3.
The authors have revised the definition of viscosity index (line 160-165). Viscosity index is defined as the integral of pressure over time from the start of injection to the end of cooling. The revision is as follows:
Viscosity index (VI): The integral of pressure over time from the start of injection to the end of cooling is primarily employed as an online quality index for the adaptive process control system and assesses changes in product weight, as shown in Equation 1.
(1) |
Where tinjection_start is the time when the start of injection, tcooling_end is the time when the end of cooling, and Pnozzle is the nozzle pressure.
Point 4.
Argumentation of adaptive control system strategy: In Section 4.2.1 of this paper, the injection speed is adjusted when the viscosity index exceeds the threshold value, and the V/P transition point is adjusted when the peak nozzle pressure exceeds the threshold value. The adjustment range is 0.1. The author should have a detailed theoretical demonstration for this adjustment relationship.
Response 4.
The authors have added detail for the adjustment relationship and modified flow chart (line 187-206). The revision is as follows:
To ensure the long-term production stability of the injection molding process, this re-search established an adaptive process control system within the MCU. The method of adaptive process control system relies on the measurement of the nozzle peak pressure and viscosity index. However, the measured values can be influenced by environmental noise, leading to potential inaccuracies. To resolve this issue, this study employed a control strategy based on average control chart theorem. In a stable production process, the nozzle peak pressure and the viscosity index should distribute within the upper control limit (UCL) and lower control limit (LCL). In each cycle, the system can calculate the characteristics of quality monitoring and automatically adjust the V/P switchover point and injection speed and monitor clamping force difference value.
The adaptive process control system adheres to the following strategy: (1) the injection speed will increase by 1 mm/s when the viscosity index exceeds the UCL; conversely, it will decrease 1 mm/s. (2) the V/P switchover point will decrease by 0.1 mm when the nozzle peak pressure exceeds the UCL; conversely, it will increase 0.1 mm. (3) consistent monitoring of the clamping force difference value approaching 0 ensures that the injection process remains stable. Figure 3 shows the control strategy of the adaptive process control system.
Figure 3. Control strategy of the adaptive process control system.
Point 5.
Limits and limits of process parameters: The range and limits of injection speeds and V/P switching points should be explicitly mentioned in the discussion. Indicate whether the variable range of these parameters is limited by equipment or other process conditions, or whether further optimization and adjustment is required.
Response 5.
In this study, the upper limit for injection speed is primarily based on the machine's maximum setting. Additionally, the lower limit for the V/P switchover point is set at 10 mm, as going below 10 mm can lead to friction between the screw and the barrel, affecting the life of machine. For the adaptive process control system, when the V/P switchover point or injection speed consistently adjusts in the same direction for 10 times, the manufacturing process will be paused. This could be attributed to issues within the system or errors in parameter settings, resulting in instability in product quality.
Point 6.
Flexibility of control strategy: Consider adding more control strategies and parameter adjustment methods to the adaptive process control system to adapt to a wider range of production conditions and process changes. This improves the flexibility and applicability of the system and further optimizes the stability of product quality.
Response 6.
In future research, the authors will consider adding more control strategies and parameter adjustment methods to the adaptive process control system. For instance, conducting experiments with different materials, incorporating thick-wall parts, and exploring various control strategies will be considered. The goal is to maintain the stability of product quality under different environmental noise, addressing a broader range of production conditions and process variations.
Point 7.
Stability and robustness analysis of the control system: It is suggested that the author should conduct the stability and robustness analysis of the developed single-chip microcomputer control system. This will help evaluate the performance of the control system under different operating and environmental conditions.
Response 7.
In future research, the authors will consider conducting a robustness analysis of the system and attempt experiments in different environments. Evaluating the system's performance under various environmental conditions and dates to enhance its applicability and generalizability.
Point 8.
Generalization of the results: Discuss the generalization and applicability of the results and analyze whether the proposed adaptive control system can be extended and applied in other injection molding processes (using different materials, molds, injection molding machines). Consider the limitations of the discussion results and any possible ways to improve them.
Response 8.
The adaptive process control system used in this study have previously been applied to hydraulic injection molding machines with molds equipped with cold runners, yielding favorable results. In this study, the system was primarily applied to all-electric injection molding machines with hot runner thin-walled molds. The results show that applying this system can effectively stabilize product weight. Currently, the pressure sensor of the system is installed at the nozzle. If applied to thick-walled molds, the control strategy of system needs to be verified for suitability. Future research directions will focus on validating these aspects in such applications.
We appreciate your kind help and hope to hear from you soon.
Sincerely,
Sheng-Jye Hwang
Professor
Department of Mechanical Engineering
National Cheng Kung University
Tainan, Taiwan

Reviewer 3 Report
Comments and Suggestions for Authors
The manusript investigate correlations between injection moulding parameters and quality of the parts, a box. An adaptive control system is introduced to improve the stability of quality (weitht of the product).
1. In the literature review, the research gap should be highlighted, what is the problem you want to solve in your research?
2. Quality of Figure 1 should be improved. And Line 135 is a wrong statement, PVT curve is about the specific volume or density of the polymer, not the weight.
3. Font for Figure 4 should be larger, especially for drawing in (a)
4. The author needs to mention, for each optimization experiments, for example the “injection speed experiments”, when the speed is varied, how were the other parameters set? Similar for other sections.
5. Figure 23, the figures don’t deliver any information, you should change the Y range or use log to make your data visible on the figure.
6. The author needs to explain the tie-bar censor’s function. Is it measuring/correlated with the clamping force? How it is correlated?
7. When adaptive system is introduced, the variation and standard deviation is slightly reduced. This variation is so low that you need to compare with production variation source (for example for different days ). Line 355 is not fully supported: that the system effectively stabilized product quality
8. Figure 19, how to define “too high/too low” in the system?
Comments on the Quality of English LanguageEnglish is ok.
Author Response
Response to Reviewer 3 Comments
<Polymers>
< Out-of-Mold Sensor Based Process Parameter Optimization and Adaptive Process Quality Control for Hot-Runner Thin-walled Injection Molded Parts>
<Manuscript ID: polymers-2850899>
Dear Reviewer,
We appreciate the time and effort the reviewer has dedicated to providing insightful feedback on ways to strengthen our paper. Thus, it is with great pleasure that we were doing our best to work on revising our article for further consideration. We have incorporated changes that reflect the detailed suggestions you have graciously provided. We also hope that our efforts and responses in revising the paper as shown below will satisfactorily address all the issues and concerns you and the reviewers have noted.
Point 1.
In the literature review, the research gap should be highlighted, what is the problem you want to solve in your research?
Response 1.
The description in Chapter 1 (line 115-121) and Chapter 4 (line 250-256) has been modified to emphasize the research gap. The description is as follows:
Based on the literature mentioned above, we observed many studies on quality monitoring and parameter optimization in injection molding. However, there is relatively little research on parameter optimization and quality monitoring for hot runner and thin-walled products, despite the widespread applications of thin-walled products. Therefore, this study utilized hot runner thin-walled parts as the experimental product, installed a pressure sensor at the nozzle to capture nozzle pressure, and a strain sensor placed on the tie bar to convert the instantaneous clamping force. (line 114-120)
Considering the widespread applications of thin-walled products, this study focused on the parameter optimization of hot runner thin-walled parts. It established a parameter optimization process suitable for thin-walled products and observing the impact of different process parameters on product quality. The procedure was optimized by establishing a standard process parameter setup through the utilization of signals from sensors installed in the machine. It can find the appropriate process parameters without the need for experienced engineers. (line 249-255)
Point 2.
Quality of Figure 1 should be improved. And Line 135 is a wrong statement, PVT curve is about the specific volume or density of the polymer, not the weight.
Response 2.
- Figure 1 has been removed because the authors consider it conveying less information.
- The description of P-V-T relationship has been modified (line 143-148). The description is as follows:
According to the P-V-T relationship, the correlation between pressure and product weight is significant. In the study, the characteristics of the nozzle pressure curve which highly correlated with product weight were defined as online quality indexes. Previous literature demonstrated the effectiveness of fitting these characteristic points for stabilizing product weight [22-23]. Furthermore, the injection molding parameters highly correlated with product weight were adjusted to establish a standard pressure curve.
Point 3.
Font for Figure 4 should be larger, especially for drawing in (a)
Response 3.
The front in Figure 4 has been enlarged, and the Figure has been replaced with clearer versions (line 237).
Point 4.
The author needs to mention, for each optimization experiments, for example the “injection speed experiments”, when the speed is varied, how were the other parameters set? Similar for other sections.
Response 4.
The authors have been revised the description of optimization experiments (line 256-263). The description is as follows:
In this research, the optimization experiment parameters were sequentially optimized. The parameter optimization process is mainly arranged according to the injection molding process. The first process parameters optimized is the injection speed, followed sequentially by V/P switchover point, packing pressure, packing time, and clamping force. The optimized parameters in each experiment were used for the following optimization experiments. After the optimization of the above parameters, a set of close-to-optimal process parameters can be obtained. The parameters for the optimization experiment are shown in Table 2.
Point 5.
Figure 23, the figures don’t deliver any information, you should change the Y range or use log to make your data visible on the figure.
Response 5.
Figure 23 shows the clamping force difference values during the adaptive process control experiments. In this study, clamping force difference values were used to monitor the separation. However, the results cannot be clearly expressed in Figure 23.
Therefore, the authors have removed Figure 23 based on the editor's suggestion. The reason for the unchanged values in Figure 23 is that the clamping force is set sufficiently, so the clamping force difference value remains at 0 throughout the adaptive process control experiments, regardless of whether the adaptive control system is used or not.
Point 6.
The author needs to explain the tie-bar sensor’s function. Is it easuring/correlated with the clamping force? How it is correlated?
Response 6.
The primary function of the tie-bar strain gauge is to measure the value of the clamping force during the injection molding process. After the mold is closed, the tie-bar strain gauge can measure the strain experienced by the tie bars. Through system conversion of this strain, the actual clamping force value is obtained. The strain sensor allows us to monitor changes in clamping force during the injection molding process to observe the occurrence of any mold separation.
Point 7.
When adaptive system is introduced, the variation and standard deviation is slightly reduced. This variation is so low that you need to compare with production variation source (for example for different days). Line 355 is not fully supported: that the system effectively stabilized product quality
Response 7.
The phenomenon of a slight reduction in variation and standard deviation is attributed to the optimization of process parameters conducted in the parameter optimization experiments, and the application of these optimized parameters in the adaptive control experiments. As the result, the product weight has become more stable. However, even with close-to-optimal parameters, product weight may still be influenced by environmental noise. Therefore, the primary objective of the adaptive process control system is to further stabilize the overall production process.
Thank you very much for your suggestion. In the future, the authors plan to conduct additional research to evaluate the performance of this system under different dates and environmental conditions. To ensure consistency in product quality under various environmental conditions.
Point 8.
Figure 19, how to define “too high/too low” in the system?
Response 8.
The authors have added the definition of “too high” and “too low” and made modifications to the Figure (line 187-206). The revision is as follows:
To ensure the long-term production stability of the injection molding process, this research established an adaptive process control system within the MCU. The method of adaptive process control system relies on the measurement of the nozzle peak pressure and viscosity index. However, the measured values can be influenced by environmental noise, leading to potential inaccuracies. To resolve this issue, this study employed a control strategy based on average control chart theorem. In a stable production process, the nozzle peak pressure and the viscosity index should distribute within the upper control limit (UCL) and lower control limit (LCL). In each cycle, the system can calculate the characteristics of quality monitoring and automatically adjust the V/P switchover point and injection speed and monitor clamping force difference value.
The adaptive process control system adheres to the following strategy: (1) the injection speed will increase by 1 mm/s when the viscosity index exceeds the UCL; conversely, it will decrease 1 mm/s. (2) the V/P switchover point will decrease by 0.1 mm when the nozzle peak pressure exceeds the UCL; conversely, it will increase 0.1 mm. (3) consistent monitoring of the clamping force difference value approaching 0 ensures that the injection process remains stable. Figure 3 shows the control strategy of the adaptive process control system.
Figure 3. Control strategy of the adaptive process control system.
We appreciate your kind help and hope to hear from you soon.
Sincerely,
Sheng-Jye Hwang
Professor
Department of Mechanical Engineering
National Cheng Kung University
Tainan, Taiwan

Round 2
Reviewer 1 Report
Comments and Suggestions for Authors
The authors responded and revised the manuscript based on the questions and comments. The control algorithm itself may be of some practical interest, but is not novel from an engineering or scientific standpoint.
Author Response
Response to Reviewer 1 Comments
<Polymers>
< Out-of-Mold Sensor Based Process Parameter Optimization and Adaptive Process Quality Control for Hot Runner Thin-walled Injection Molded Parts >
<Manuscript ID: polymers-2850899>
Dear Reviewer,
We appreciate the time and effort the reviewer has dedicated to providing insightful feedback on ways to strengthen our paper. Thus, it is with great pleasure that we were doing our best to work on revising our article for further consideration. We have incorporated changes that reflect the detailed suggestions you have graciously provided. We also hope that our efforts and responses in revising the paper as shown below will satisfactorily address all the issues and concerns you and the reviewers have noted.
Point 1.
The control algorithm itself may be of some practical interest, but is not novel from an engineering or scientific standpoint.
Response 1.
This study installed a pressure sensor at the nozzle to monitor the injection molding process, optimizing injection molding parameters, and stabilizing the product weight under reasonable cost for hot runner thin-walled parts. The typical pressure profile for hot runner thin-walled parts is very different from those of regular thickness. Thus, the authors have put lots of efforts in analyzing and using the nozzle pressure profile to develop the control algorithm for the process parameter setup and the adaptive process control system. This paper also demonstrate that this process parameter setup process and adaptive control system can be applied to all-electric injection molding machines. The experiment results also explain and archive the phenomenon of melt nozzle pressure profile at high injection speeds during thin-walled product processing. These results will benefit and set foundation for future research of thin-walled injection molding process.
We appreciate your kind help and hope to hear from you soon.
Sincerely,
Sheng-Jye Hwang
Professor
Department of Mechanical Engineering
National Cheng Kung University
Tainan, Taiwan

Reviewer 3 Report
Comments and Suggestions for Authors
The PVT curve: it is true all the parameters will influence the weight, however, on the PVT curve, it is the specific volume as the Y axis.
Comments on the Quality of English LanguageIt is ok.
Author Response
Response to Reviewer 3 Comments
<Polymers>
< Out-of-Mold Sensor Based Process Parameter Optimization and Adaptive Process Quality Control for Hot Runner Thin-walled Injection Molded Parts >
<Manuscript ID: polymers-2850899>
Dear Reviewer,
We appreciate the time and effort the reviewer has dedicated to providing insightful feedback on ways to strengthen our paper. Thus, it is with great pleasure that we were doing our best to work on revising our article for further consideration. We have incorporated changes that reflect the detailed suggestions you have graciously provided. We also hope that our efforts and responses in revising the paper as shown below will satisfactorily address all the issues and concerns you and the reviewers have noted.
Point 1.
The PVT curve: it is true all the parameters will influence the weight, however, on the PVT curve, it is the specific volume as the Y axis.
Response 1.
The reciprocal of specific volume is density. Thus, P-V-T relationship is used to follow the density of a polymer during processing, and it can be used to identify weight of a part during injection molding process. The authors have modified the description of P-V-T relationship (line 131-148). The description is as follows:
2.1 P-V-T relationship
The P-V-T relationship refers to the interdependence of pressure (P), specific volume (V), and temperature (T) for polymers. Under constant temperature, increasing the pressure will decrease the specific volume or increase density. Conversely, under constant pressure, increasing the temperature will increase the specific volume or decrease the density. The final specific volume after cooling significantly will affect product quality, including dimensions, mechanical properties, product weight, etc.
According to the P-V-T relationship, the pressure generated in the injection process will affect the specific volume or density and directly affect the product weight. Therefore, setting appropriate injection molding parameters to maintain similar pressure curve for each cycle, and the stability of the product weight will increase.
2.2 Characteristics of the nozzle pressure profile
According to the P-V-T relationship, the correlation between pressure and product weight is significant. In the study, the characteristics of the nozzle pressure curve which highly correlated with product weight were defined as online quality indexes. Previous literature demonstrated the effectiveness of fitting these characteristic points for stabilizing product weight [22-23]. Furthermore, the injection molding parameters highly correlated with product weight were adjusted to establish a standard pressure curve.
We appreciate your kind help and hope to hear from you soon.
Sincerely,
Sheng-Jye Hwang
Professor
Department of Mechanical Engineering
National Cheng Kung University
Tainan, Taiwan
